# Significance of Singlet Oxygen Molecule in Pathologies

**DOI:** 10.3390/ijms24032739

**Published:** 2023-02-01

**Authors:** Kazutoshi Murotomi, Aya Umeno, Mototada Shichiri, Masaki Tanito, Yasukazu Yoshida

**Affiliations:** 1Biomedical Research Institute, National Institute of Advanced Industrial Science and Technology (AIST), Tsukuba 305-8566, Japan; 2Department of Ophthalmology, Shimane University Faculty of Medicine, Izumo 693-8501, Japan; 3Biomedical Research Institute, National Institute of Advanced Industrial Science and Technology (AIST), Ikeda 563-8577, Japan; 4LG Japan Lab Inc., Yokohama 220-0011, Japan

**Keywords:** reactive oxygen species, singlet oxygen, biomarkers, lipid peroxidation

## Abstract

Reactive oxygen species, including singlet oxygen, play an important role in the onset and progression of disease, as well as in aging. Singlet oxygen can be formed non-enzymatically by chemical, photochemical, and electron transfer reactions, or as a byproduct of endogenous enzymatic reactions in phagocytosis during inflammation. The imbalance of antioxidant enzymes and antioxidant networks with the generation of singlet oxygen increases oxidative stress, resulting in the undesirable oxidation and modification of biomolecules, such as proteins, DNA, and lipids. This review describes the molecular mechanisms of singlet oxygen production in vivo and methods for the evaluation of damage induced by singlet oxygen. The involvement of singlet oxygen in the pathogenesis of skin and eye diseases is also discussed from the biomolecular perspective. We also present our findings on lipid oxidation products derived from singlet oxygen-mediated oxidation in glaucoma, early diabetes patients, and a mouse model of bronchial asthma. Even in these diseases, oxidation products due to singlet oxygen have not been measured clinically. This review discusses their potential as biomarkers for diagnosis. Recent developments in singlet oxygen scavengers such as carotenoids, which can be utilized to prevent the onset and progression of disease, are also described.

## 1. Introduction

Oxygen can be recognized as a “double-edged sword” because aerobic organisms cannot live without it, even though the oxidative metabolism produces toxic by-products. Most deleterious phenomena arise because of free radicals. Hence, for survival, the elimination of highly reactive oxygen species (ROS) is important for living organisms. The human body is also constantly threatened by oxidative injury and radicals. Over the course of evolution, humans have developed defense systems to counteract oxidative damage, including the transformation and/or elimination of ROS, as a part of normal physiological homeostasis. When this homeostatic balance is disturbed by stress caused by environmental or radiation exposure, it is termed oxidative stress, that is, a state in which the homeostatic balance between oxidation reactions and antioxidant defenses is lost, leading to the oxidation of vital biomolecules and, finally, the onset of disease.

Thousands of studies have been reported over the past decade on the role of ROS in cell and tissue injury. ROS are generally accepted to be involved in various pathologies, including inflammation, asthma, muscular dystrophy, dementia, anaphylaxis, rheumatoid arthritis, reperfusion injury following ischemic stroke or heart attacks, cardiac toxicity of anti-cancer drugs, carcinogenicity of various chemicals, and smoking [1,2,3,4,5].

In this review, we focus on singlet oxygen (^1^O_2_) in ROS. However, we exclude photodynamic therapy (PDT), in which ^1^O_2_ is stimulated by the light irradiation of a photosensitizing agent to induce a therapeutic effect on cancer [6], because many interesting review articles on PDT have already been published in recent years. The molecular aspects of (1) the mechanism of ^1^O_2_ production in vivo, (2) methods for detecting ^1^O_2_ and its oxidation-modification products, (3) implications in the pathogenesis of skin and eye diseases, and (4) compounds that scavenge ^1^O_2_ are presented. Furthermore, we present our findings on the measurement of ^1^O_2_-mediated oxidation products in diabetes, bronchial asthma, and ocular diseases. Although ^1^O_2_ has been known to potentially be involved in the pathogenesis of certain diseases, lipid peroxidation products and other products produced by ^1^O_2_ have not been actively measured clinically. In this review, we examine the usefulness of lipid peroxidation products produced by ^1^O_2_ in diagnosing and understanding the pathophysiology of diseases.

## 2. Chemical Properties and Possible Production of Singlet Oxygen In Vivo

ROS are redox-active intermediates that are formed by the chemical, photochemical, or biochemical reduction in oxygen that, at least partially, triggers a chain of oxidative reactions. ROS are grouped into non-radical and radical species, the former including ^1^O_2_ and hydrogen peroxide (H_2_O_2_), and the latter including superoxide anions (O_2_^•−^), hydroxyl radicals (HO^•^), hydroperoxyl radicals (HO_2_^•^), alkoxyl radicals (RO^•^), and peroxyl radicals (ROO^•^) [7]. ^1^O_2_ is generated by energy transfer via chemical or photochemical pathways from an activated species to the oxygen molecule (triplet oxygen, ^3^O_2_) (Figure 1). Other species are formed by a series of reactions initiated by ^1^O_2_, or by certain one-electron reduction processes [7]. Therefore, ROS have been actively studied over the past three decades to determine whether they are beneficial or harmful.

High-energy radiations, such as X- and γ-rays, generate radical ROS directly by the ionization of H_2_O or oxygen, whereas various chemical and photochemical reactions produce ^1^O_2_ (Figure 2A), which in turn creates other ROS [8]. In the energy transfer mechanism, a photoactivated chromophore causes intersystem crossing to the triplet state, and triplet–triplet annihilation transfers part of the electronic energy to oxygen to generate ^1^O_2_ [9,10]. Since the energy of most photoactivated molecules is higher than the excitation energy of ^1^O_2_ (0.98 eV) (Figure 1), ^1^O_2_ is readily produced by endogenous chromophores activated by ultraviolet (UV) irradiation [11]. Hatz et al. showed that the lifetime of ^1^O_2_ generated by pulsed laser irradiation of a photosensitizer (5,10,15,20-tetrakis(N-methyl-4-pyridyl)-21H, 23H-phine (TMPyP)) incorporated into the nucleus of HeLa cells is about 3 μs. They also found that the generated ^1^O_2_ diffuses from the formation point to a sphere radius of about 100 nm [12,13]. On the other hand, Liang et al. created a system that generates organelle-specific ^1^O_2_ by irradiating 660 nm light using a photosensitizer that is locally expressed in the membrane, cytosol, endoplasmic reticulum, mitochondria, and nucleus [14]. Using this system, they observed differences in the irradiation energy required to induce cell death depending on the organelle in which the photosensitizer is expressed, as well as differences in the type of cell death (early apoptosis, necrosis, or late apoptosis) [14]. This result indicates that ^1^O_2_ generated in intracellular organelles may not spread to other organelles. These reports indicate that ^1^O_2_ generated by intracellular photosensitizers may induce cell death by oxidizing the components of the organelle containing the photosensitizer, rather than diffusing widely within the cell.

The secretion of myeloperoxidase (MPO) from phagocytes generates hypochlorite (ClO^−^), which reacts with H_2_O_2_ to form ^1^O_2_ (Figure 2B) [15]. In leukocytes, ^1^O_2_ generated via peroxidases and nicotinamide adenine dinucleotide phosphate (NADPH) oxidase critically contributes to an antimicrobial role [16]. Another endogenous pathway of ^1^O_2_ is the degradation of lipid peroxides or ROO^•^ via the Russel mechanism (Figure 2C) [17,18]. ROO^•^ and lipid peroxides, which are the substrates of the Russel mechanism, are derived from the peroxidation of polyunsaturated fatty acids (PUFA) by auto-oxidation and enzymatic-oxidation, such as cytochrome c, lactoperoxidase [19], and lipoxygenases [20]. In addition, the light-independent generation pathways of ^1^O_2_ include reactions involving O_2_^•−^ (Figure 2D) [21], peroxynitrite (Figure 2E) [22,23], and ozone [24,25,26]. Atmospheric particulates also catalyze the ^1^O_2_ generation [27,28].

## 3. Damage to Biomolecules by ^1^O_2_-Mediated Oxidation

Notably, because ^1^O_2_ has electrophilic properties, it attacks the π-bonds of the compound to form hydroperoxide by the ene reaction or endoperoxide by the 1,4-addition reaction (Figure 3A). ^1^O_2_ oxidatively modifies biomolecules, such as amino acids (Figure 3B) [29,30], nucleic acids (Figure 3C) [31,32], and lipids (Figure 3D) [33,34], either by a direct reaction or by the induction of ROS. 

Amino acids, which are components of proteins, are the targets of ^1^O_2_. Among the 20 natural amino acids, tryptophan, histidine, tyrosine, and the sulfur-containing amino acids cysteine and methionine, are susceptible to reaction with ^1^O_2_ because of their structural properties. The reaction products of tryptophan and methionine with ^1^O_2_ are shown in Figure 3B. Tryptophan reacts with ^1^O_2_ to form a hydroperoxide at the C3 position as an intermediate, which undergoes ring closure to form hydroxypyrroloindole, and finally, N-formylkynurenine [35]. Tryptophan reacts with ^1^O_2_ to form C2-C3 cross-linked dioxetanes as intermediates, followed by the conversion to N-formylkynurenine [35]. Methionine reacts with ^1^O_2_ to form methionine sulfoxide and the cyclic product dehydromethionine via a persulfoxide intermediate [30]. Oxidative modifications to amino acids cause alterations in protein properties, resulting in cellular damage accompanied by reduced enzyme activity and disruption of the cell structure.

^1^O_2_ reacts specifically with the guanine portion of DNA bases converting to 7,8-dihydro-8-oxo-2′-deoxy-guanin (8-oxo-dG) (Figure 3C) [36]. 8-oxo-dG is a potential mutagenic substance because it pairs with adenine to cause a G → T transversion, and it is widely recognized as one of the indicators for evaluating oxidative DNA damage [37]. The details are discussed below in Section 5.1.4. on skin cancer.

Phospholipids in the membrane react with ^1^O_2_, resulting in the oxidative modification of fatty acids (Figure 3D). Oxidized fatty acids are cleaved and serve as signaling molecules involved in carcinogenesis and skin aging. The cleavage of short-chain aldehydes from fatty acids also results in membrane instability and increased permeability. The oxidation products derived from linoleic acid and cholesterol by ^1^O_2_ are described below.

^1^O_2_ also oxidatively modifies steroids [38], vitamins [39,40], carbohydrates, terpenes [41], and flavonoids [42]. In addition, ^1^O_2_ contributes to the stimulation of stress-activated kinases and the regulation of gene expression [43].

## 4. Detection Methods of ^1^O_2_ In Vivo and In Vitro

The oxidative modification of biomolecules by ^1^O_2_ has been related in several diseases, including skin and eye diseases [44]. The development of methods to precisely detect and quantify ^1^O_2_ contributes not only to an understanding of its roles in physiological and pathological conditions, but also in cancer therapy, by understanding the mechanism of PDT. Detection methods for ^1^O_2_ have been developed to verify ^1^O_2_ production in living systems [45]. The detection of ^1^O_2_ is mainly performed using the following methods: spectroscopic measurement of near-infrared (NIR) luminescence [46,47], electron spin resonance (ESR) with sterically hindered amine [48,49], fluorescence measurement by reaction with fluorescence probes [50,51], and the measurement of oxidation products produced by the reaction of ^1^O_2_ with biomolecules [52,53]. These methods were utilized for in vitro and in vivo experiments. 

### 4.1. Direct Detection of ^1^O_2_

The measurement of ^1^O_2_ NIR luminescence is a reliable method for the direct spectroscopic detection of ^1^O_2_, as ^1^O_2_ emission (1270 nm) can be excited by an argon laser [54]. This luminescence emission was first used for the time-resolved detection of ^1^O_2_ in solution by Krasnovsky in 1976. This measurement method has been used as a standard technique for ^1^O_2_ formation yields, lifetimes, and deactivation constants in various solutions. The in vivo detection of ^1^O_2_ luminescence has been attempted previously, and it has been reported that ^1^O_2_ in a suspension of leukemia cells [55,56] and red cell ghost [57] can be detected using a sophisticated near-infrared photomultiplier device. However, this luminescence signal is weak because the inactivation of ^1^O_2_ is dominated by nonradiative pathways; typical phosphorescence yields are of the order of 10^−5^ to 10^−7^ [58]. In addition, these methods require the usage of deuterium oxide (D_2_O) to extend the lifetime of ^1^O_2_ and remove the 1270 nm emission absorption due to H_2_O [59]. This difficulty in detecting ^1^O_2_ in vivo can be attributed to the short lifetime of ^1^O_2_ in cells and tissues and the lack of sufficient sensitive detectors at NIR wavelengths. 

ESR spectroscopy, which has been proposed for paramagnetic materials and organic compounds with unpaired electrons, has been used for the detection of free radicals [60]. To detect ^1^O_2_, a method was developed for measuring nitroxide radicals produced by the reaction of ^1^O_2_ with sterically hindered secondary amine probes such as 2,2,6,6-tetramethylpiperidine (TEMP) (Figure 4A) [61,62,63] and hydroxy-TEMP (Figure 4B) [64]. This method has been applied to the in vitro ^1^O_2_ scavenging activity of various substances in solution, and there is considerable evidence demonstrating the ^1^O_2_ scavenging effects of various substances using this method [48,65,66]. However, ESR spectroscopy is not considered suitable for the detection of intracellular ^1^O_2_, which may be because the time resolution in ESR measurements is not suitable for the short lifetime of ^1^O_2_ in cells existing in ^1^O_2_ quenching molecules.

### 4.2. ^1^O_2_ Detection by Fluorescent Probes

Fluorescent probes are excellent sensors for the detection of ROS because of their high sensitivity, simplicity of data acquisition, and high spatial resolution in microscopic imaging [67,68,69]. Fluorescent probes, including 9,10-dimethylanthracene (DMA) (Figure 4C) [68,70,71] and 9-[2-(3-Carboxy-9,10-diphenyl)anthryl]-6-hydroxy-3H-xanthen-3-ones (DPAXs) (Figure 4D) [72], have been developed for the detection of ^1^O_2_ in neutral or basic aqueous solutions. These probes react specifically and rapidly with ^1^O_2_ to form stable endoperoxides with a high rate constant. Singlet Oxygen Sensor Green (SOSG) (Figure 4E) is a fluorescent probe for in vitro detection because of its high ^1^O_2_ selectivity [73,74,75]. To apply fluorescence probes in biological samples, the probes must penetrate the cell membrane and be localized in the cell. Recently, a far-red fluorescent probe consisting of DMA and silicon-containing rhodamine (Si-rhodamine) moieties, namely Si-DMA (Figure 4F), was developed for detecting ^1^O_2_ in the mitochondria. The advantages of Si-DMA are its cell-permeable ability and increased sensitivity to specifically detect mitochondrial ^1^O_2_ by the Si-rhodamine moiety [76]. It is important to precisely measure intracellular ^1^O_2_ generation to clarify the characteristics of fluorescence probes in terms of their signal-to-noise ratio and dynamic changes. We have demonstrated that intracellular ^1^O_2_ can be quantitatively measured using Si-DMA in living cells, as time-lapse imaging using Si-DMA provides an apparent signal-to-noise ratio in the treatments of a ^1^O_2_ generator and quencher [77]. Other fluorescent probes have been used to detect intracellular ^1^O_2_ using nanoparticles, including SOSG-based nanoprobes [50], super-pH-resolved nanosensors encoding SOSG [78], and biocompatible polymeric nanosensors encapsulating SOSG within their hydrophobic core [79]. The use of these fluorescence probes will improve our understanding of the ^1^O_2_ generation mechanism and the biological function of ^1^O_2_ in the physiological and pathological conditions of cultured cells.

### 4.3. Detection of ^1^O_2_-Mediated Peroxidation Products

^1^O_2_ generated in vivo quickly reacts with biomolecules, including lipids, proteins, and nuclei, and the subsequent comparatively stable oxidized products remain in the ^1^O_2_ generation site. This chemical property has been used for the development of in vivo ROS detection methods, and these oxidation products are widely used as oxidative stress biomarkers because they can be detected stably and correlate with oxidative stress status in vivo [80,81].

The oxidation product of linoleic acid, hydroxyoctadecadienoic acid (HODE), contains six isomers. ^1^O_2_ reacts with molecules containing double bonds to form hydroperoxides as primary products, which are subsequently reduced to hydroxides (Figure 5). ^1^O_2_ produces 9-, 10-, 12-, and 13-(*Z*,*E*)-hydroperoxyoctadecadienoic acid (HPODE) from linoleates; however, only 10- and 12-(*Z*,*E*)-HPODE are specific products by ^1^O_2_ because 9- and 13-(*Z*,*E*)-HPODE are also formed by both free radical- and enzyme-mediated oxidation. The chemical structures of 10- and 12-(*Z*,*E*)-HPODE differ significantly from those of 9- and 13-(*Z*,*E*)-HPODE: the latter contains a conjugated diene, while the former does not. These structural features cause differences in physiological functions; for example, 10- and 12-(*Z*,*E*)-HODE cause adaptive responses to UV-derived oxidative damage in cultured cells, whereas 9- and 13-(*Z*,*E*)-HODE do not [82,83]. 

Cholesterol is an important lipid that constitutes biological membranes and undergoes oxidative modification to produce 7-, 24-, and 27-hydoroxycholesterol. Cholesterol, as well as unsaturated fatty acids, are substrates for the oxidative reaction of ^1^O_2_. When ^1^O_2_ reacts with cholesterol, it forms 5α-hydroperoxide (cholesterol 5α-OOH), 6α-hydroperoxide (cholesterol 6α-OOH), and 6β-hydroperoxide (cholesterol 6β-OOH) (Figure 6). Cholesterol 5α-OOH is produced in larger amounts than 6α/β-OOH. Therefore, cholesterol 5α-OOH could be used as a biomarker for oxidative reactions involving ^1^O_2_ in vivo. Furthermore, the Hock cleavage of cholesterol 5α-OOH converts to cholesterol 5,6-secosterol. Cholesterol 5,6-secosterol, also called ateronal, is involved in the development of cardiovascular diseases [84] and neurodegeneration [85]. 

We have developed a method to comprehensively measure lipid peroxidation and proposed the measurement of HODEs and hydroxycholesterol [86,87,88,89,90,91]. In this method, lipid components are extracted from biological samples after reduction and saponification. These pretreatments result in the measurement of both free and esterified forms of hydroperoxides and hydroxides as free hydroxides. HODEs and hydroxycholesterol are analyzed by liquid chromatography–tandem mass spectrometry (LC-MS/MS) and gas chromatography–mass spectrometry (GC-MS), respectively. Biological samples to be measured with this method include plasma, urine, erythrocytes, as well as cultured cells and tissues. This method allows for the measurement of not only radical-mediated oxidation products, but also ^1^O_2_-specific oxidation products using the HODEs isomer as an indicator [89,90,91], in other words, indirect in vivo ^1^O_2_ detection is possible. We have determined the relationship between ^1^O_2_-mediated oxidation and the early diagnosis of diabetes in humans and mice based on the production of 10- and 12-(*Z*,*E*)-HODEs [83,92,93], which will be discussed in more detail later. ^1^O_2_ measurements using 10- and 12-(*Z*,*E*)-HODEs as indicators can evaluate ^1^O_2_ generation not only in living cells, but also in humans and animals in vivo, and they can precisely analyze the involvement of ^1^O_2_ generation and diseases.

While ESR and fluorescent probes are useful for detecting ^1^O_2_ in vitro and in cells, it is difficult to use these tools to assess the degree of damage caused by ^1^O_2_ in tissues. Therefore, it is better to measure oxidation products that are relatively stable and retained in biological samples. Table 1 summarizes the methods for measuring ^1^O_2_ itself or its oxidation products and both their advantages and disadvantages. Table 2 summarizes reports of the measurements of linoleic acid- or cholesterol-derived oxidation products produced by ^1^O_2_-mediated oxidation reactions in human and animal samples. ^1^O_2_-derived lipid oxidation products have been measured in the blood and tissue samples obtained from patients with borderline diabetes, glaucoma, alcoholism, and atherosclerosis. In animal experiments, cholesterol 5α-hydroperoxide in the skin tissue was measured following UVA irradiation in mice. These results are important to understand the contribution of ^1^O_2_ to the pathogenesis of various disorders.

## 5. Diseases Involving ^1^O_2_ and Their Pathophysiology

### 5.1. Skin Diseases

The skin is the organ with the largest area exposed to sunlight. The skin is regularly exposed to sunlight and is at a high risk of oxidative stress. UV radiation from the sun generates ROS in keratinocytes, the major cells of the epidermis and the outermost layer of the skin [105]. UV exposure also increases the activities of cyclooxygenase and lipoxygenase, which produce eicosanoids from arachidonic acid. UV exposure of the skin causes erythema and acute inflammation, and long-term UV exposure causes photo-carcinogenesis [106] and skin aging [107]. Lipid oxidation on the skin surface is important in relation to sunburn, hyperpigmentation, wrinkle formation, freckles, atopic dermatitis, acne, and skin cancer.

UVA (320–400 nm) penetrates the dermis, whereas UVB (280–320 nm) only attacks the epidermis. Both UVA and UVB produce ROS and free radicals in the presence of photosensitizers (Figure 7). In type I reactions, an excited photosensitizer reacts with an organic compound to produce O_2_^•−^ and lipid peroxyl radical (LOO^•^). In a type II reaction, the excited photosensitizer reacts with a triplet oxygen molecule, producing ^1^O_2_ primarily by energy transfer and, to a minor extent, O_2_^•−^ by electron transfer.

#### 5.1.1. Photosensitizers

Porphyrins such as protoporphyrin IX (Figure 8A) [108] have been examined as endogenous photosensitizers, and their derivatives [8] and 5-aminolevulinic acid [109], a component of porphyrins, have been investigated for application as PDT in therapeutic modalities against diseases such as cancer. Interestingly, coproporphyrin (Figure 8B) produced by *Propionibacterium acnes,* an acne-causing bacterium, also acts as a photosensitizer and generates ^1^O_2_, which is associated with skin inflammation [110,111]. Hemin (Figure 8C) [112] and chlorophyll (Figure 8D) [113], which are metal complexes of porphyrin and similar compounds, also exhibit photosensitizing effects. Hemin is an iron-containing protoporphyrin IX, and chlorophyll is a chemical responsible for absorbing light energy in the light reaction of photosynthesis. Pheophytin (Figure 8E), from which magnesium ions are removed during chlorophyll degradation, and pheophorbide (Figure 8F), from which the phytyl group is eliminated from pheophytin, also act as photosensitizers [114,115]. 

In addition to porphyrins, tryptophan and tryptophan metabolites (kynurenine and 3-hydroxykynurenine) (Figure 8G) [116] and riboflavin (Figure 8H) [117,118] are endogenous photosensitizers. Riboflavin, a water-soluble vitamin B2, is distributed in the skin, eyes, brain, and blood vessels and increases the production of ^1^O_2_. Cholesta-5,7,9(11)-trien-3beta-ol (9-DDHC) (Figure 8I), a metabolite of 7-dehydrocholesterol (7-DHC), is a photosensitizer involved in rare cholesterol metabolism disorders [119]. Psoralen (Figure 8J), found in citrus fruits, is a dietary photosensitizer that causes photosensitivity [120,121].

#### 5.1.2. Lipids in the Skin and Lipid Peroxidation Products Mediated by ^1^O_2_

The lipid composition of the epidermis is a mixture of free fatty acids, ceramide, and cholesterol at approximately 30–35% each [122]. The fatty acids in the epidermis include linoleic acid (21%), oleic acid (15%), palmitic acid (14%), stearic acid (11%), and arachidonic acid (6%) [123]. In contrast, the lipid composition of sebum is triacylglycerol (45–50%), wax ester (25%), squalene (Sq) (12%), and free fatty acids (10%) [122]. The fatty acid composition of sebum is palmitic acid (22%), palmitoleic acid (21%), oleic acid (15%), myristic acid (12%), and myristoleic acid (5%) [124]. The lipid composition of the sebum differs significantly from that of the epidermis.

Squalene (Sq) is a triterpene compound characteristically present in sebum, and six Sq monohydroperoxides (2-, 3-, 6-, 7-, 10-, and 11-OOH-Sq) are produced in similar amounts by the peroxidation of ^1^O_2_ (Figure 9) [125]. In contrast, 2- and 3-OOH-Sq were recently reported to be produced by free radical oxidation [125]. Considering these results, 6-, 7-, 10-, and 11-OOH-Sq may be the specific products of ^1^O_2_. SqOOHs are also associated with wrinkle formation [126] and inflammatory acne [127]. As SqOOHs are unstable, further photooxidation was revealed to produce 2-OOH-3-(1,2-dioxane)-Sq (Figure 9) as a secondary oxidation product [128]. SqOOH concentrations in sebum collected from skin exposed to tobacco heating products or electronic cigarette aerosols are lower than that in sebum from skin exposed to tobacco smoke, indicating consumer hygiene and cosmetic benefits [129]. 

Table 3 shows the results of SqOOH measurements in human and animal sebum samples. The fact that SqOOHs, which are specifically produced by ^1^O_2_, can be detected using sebum, which can be collected without severe invasion, makes it possible to assess the effects of pollutants, sunlight, and UV. In the future, techniques for the measurement of the SqOOH content in sebum could be further developed in the field of skin diseases, including cosmetology.

As mentioned above, ^1^O_2_ peroxidation products from cholesterol are caused by ene reactions to 5,6 double bonds to form cholesterol 5α-OOH, 6α-OOH, and 6β-OOH (Figure 6). However, cholesterol 5α-OOH is more abundant than cholesterol 6α- and 6β-OOH [52].

PUFAs in the skin are also target molecules for peroxidation by ^1^O_2_. UVA irradiation of hairless mice has also been reported to increase 10-HPODE and 12-HPODE levels (Figure 5), which are ^1^O_2_-specific lipid hydroperoxides of linoleic acid [130].

**Table 3 ijms-24-02739-t003:** Report on the results of the determination of squalene-derived peroxides produced by oxidation reactions mediated by ^1^O_2_ using human and animal samples.

^1^O_2_-Mediated Oxidation Product	Sample Species	Disease/Model	Ref.
Squalene monohydroperoxides	Human	Increased by exposure of human skin to tobacco smoke	[131]
	Human	Sunlight exposure increases SqOOH in sebum	[132]
	Human	UV irradiation to human sebum	[133,134,135]
	Human	SqOOH reduction in sebum by application of cosmetics	[136]
	Human	Human skin surface lipids	[137]
	Human	Increased SqOOH in the scalp due to high dandruff	[138]
	Human	Detection using human fingerprints	[139]
	Human	Increase in SqOOH in the sebum due to ambient dust and ozone	[140]
	Human	Less SqOOH in sebum exposed to tobacco heating products and electronic cigarettes compared to cigarette smoke exposure	[129]
	Pig	Cigarette smoke exposure to pig skin	[141]
	Rabbit	UVA irradiation of rabbit ears	[127]
	Guinea pig	Carotene reduces the increase in SqOOH caused by UV irradiation of the skin	[142]
2-OOH-3-(1,2-dioxane)-squalene	Human	Identification of cyclic peroxides of SqOOH	[128]

#### 5.1.3. Skin Aging

The main clinical sign of skin aging is the dysfunction of the dermis, with ultrastructural disruption of elastic fibers [143]. This is due to an imbalance in collagen remodeling relative to proteolysis that occurs in the extracellular matrix. UVA-irradiated cultured human fibroblasts and human epidermis have been shown to induce the expression of collagenase, a protease that degrades elastic fibers in the dermis [144]. Moreover, the addition of cholesterol oxidants derived from the ^1^O_2_ oxidation reaction to mouse fibroblasts causes them to accumulate in lipid rafts, as well as upregulates matrix metalloprotease-9 (MMP-9) activity [145]. The UVA irradiation of cells under protoporphyrin administration results in the upregulation of MMP-9 activity with the formation of cholesterol oxide [145]. This suggests that cholesterol oxidation products produced by ^1^O_2_ may alter the structure of lipid rafts, resulting in the activation of signaling pathways that induce MMP-9 expression, leading to skin aging.

#### 5.1.4. Skin Cancer

DNA nucleobases are targets of oxidation reactions [44]. ^1^O_2_, as well as highly reactive HO^•^, are oxidants that can damage cellular DNA [146,147,148]. Since UVA is among the carcinogens to which organisms are exposed, it is important to understand the mechanisms that cause DNA damage [149]. UVA in sunlight is a potential source of oxidative DNA damage in the skin [150,151]. As discussed in the previous chapter, exposing skin to UVA radiation produces ^1^O_2_. The reactivity of ^1^O_2_ to nucleobases tends to occur in the order guanine > cytosine > adenine > uracil > thymine [152].

The introduction of an oxo group into the C8 of guanine and the addition of a hydrogen atom to the nitrogen of N7 yields 8-oxo-dG (Figure 3C). The presence of 8-oxo-dG in DNA causes problems in the S phase of the cell cycle. In the S phase, cells must replicate an exact copy of the genome. Unlike most other types of DNA damage, 8-oxo-dG does not stop the replication process but rather creates a point mutation. Normally, guanine pairs with cytosine (C), whereas 8-oxo-dG mimics thymine (T), thus forming a pair with adenine (A) [153]. Furthermore, the A:8-oxo-dG mispair does not cause helix distortion in the DNA backbone, thus avoiding correction by the error detection mechanism of replication polymerase [154]. This results in mutations that are essentially C:G→A:T mutations. The presence of C:G→A:T mutations in many cancers emphasizes the importance of 8-oxo-dG in cancer pathogenesis [155]. The amount of 8-oxo-dG produced is approximately 10^3^/cell per day in normal cells, but increases to 10^5^/cell in cancer cells [156]. 8-oxo-dG is also markedly increased in human skin cells after UVA irradiation [157]. There are reports that 8-oxo-dG is involved in photocarcinogenesis [158,159]. It has also been reported that 8-oxo-dG is increased in epidermal cells after chronic broadband UVB irradiation in Ogg1 knockout mice, which are unable to eliminate oxidized bases and are susceptible to skin cancer [160]. However, cyclobutane pyrimidine dimers observed in UV-irradiated DNA are also known to correlate with skin cancer development [161]. It is thought that DNA mutation due to ^1^O_2_ generation is not the only cause of UV-induced carcinogenesis, but that various mutagenic molecules are involved in the pathogenesis in a complex manner.

Furthermore, 8-oxo-dG has been used as a biomarker to evaluate the degree of oxidative stress, and its application has been attempted as a risk assessment for many diseases [162].

#### 5.1.5. Porphyria

Porphyria is a rare genetic disorder caused by a deficiency in any of the eight enzymes involved in the heme metabolism, resulting in the accumulation of the photosensitizing substance porphyrin or its precursors [163]. Porphyrin accumulation in the skin also causes photosensitivity, resulting in burn-like skin lesions [164]. The nine porphyria are divided into “cutaneous porphyria” and “acute porphyria”. Among cutaneous porphyria, photosensitivity symptoms begin to appear shortly after birth for congenital erythropoietic porphyria (CEP) and hepato-erythropoietic porphyria (HEP), around age 5–6 for erythropoietic protoporphyria (EPP), and after age 50 for late-onset porphyria (PCT). Exposure to sunlight causes pain, itching, redness, and swelling, and, in severe cases, blister-like burns. Porphyrins accumulated in skin tissue become excited by the absorption of light and induce the production of ^1^O_2_, resulting in tissue and vascular damage due to complement activation. In addition, the release of histamine, kinins, and chemotactic factors is thought to cause skin damage [165]. Currently, there is no cure for porphyria, and photoprotection is the only way to treat this disease. Beta-carotene, a scavenger of ^1^O_2_, is widely used to treat photosensitivity caused by EPP [166,167], but it has also been reported to be less useful for photosensitivity associated with PCT [167,168].

The ^1^O_2_ produced via porphyrins causes protein oxidation and aggregation [169,170]. In a porphyria model mouse, in which porphyrin accumulation was induced by 3,5-diethoxycarbonyl-1,4-dihydrocollidine, protein aggregation in liver tissue was observed upon safelight exposure compared to non-exposure [171]. Moreover, overexposure to sunlight in EPP and PCT is associated with liver dysfunction, cirrhosis, gallstones, and acute liver failure, and the relationship between hepatic damage in porphyria and ^1^O_2_ production by irradiation is becoming clearer.

#### 5.1.6. Smith–Lemli–Opitz Syndrome and Statin-Induced Skin Disorders

Smith–Lemli–Opitz syndrome (SLOS) is caused by mutations in the *DHCR7* gene, which encodes 7-dehydrocholesterol (7-DHC) reductase, involved in the final step of cholesterol biosynthesis [172,173]. SLOS shows a cholesterol deficiency and increased 7-DHC in the plasma and tissues [173]. A variety of symptoms are recognized in SLOS, including growth retardation, microcephaly, intellectual disability, characteristic facial features, and external malformations; however, hypersensitivity to ultraviolet light is recognized as a symptom of SLOS [174]. Although 7-DHC itself does not absorb UVA and is not a direct source of photosensitivity in SLOS, 9-DDHC (Figure 8I), a 7-DHC metabolite found in the plasma of SLOS patients, is highly absorbable to UVA [119]. Furthermore, UVA exposure to 9-DDHC generates ^1^O_2_, providing a mechanism for the pathogenesis of photosensitivity in SLOS [119]. 

In addition, some statins inhibit 7-DHC reductase [175]; thus, the accumulation of 9-DDHC may also be involved in statin-induced skin photosensitivity [176] and cataracts [177].

#### 5.1.7. Skin Disorders Caused by Exogenous Photosensitizers

Pheophorbide (Figure 8F), a degradation product of chlorophyll (Figure 8D), is a photosensitizing substance [178]. Pheophorbide is sometimes found in food, and the exposure of humans [179] and animals [180] to light after ingesting this compound can cause dermatitis due to photosensitivity. Cholesterol 5α-OOH (Figure 6) was detected in rat skin administered with pheophorbide and exposed to visible light [99].

The thiopurine prodrugs azathioprine and 6-mercaptopurine are widely prescribed for the treatment of leukemia and autoimmune diseases [181]. Although these drugs have shown therapeutic efficacy, long-term administration has been reported to increase the risk of basal cell carcinoma and squamous cell carcinoma by 10- and 65- to 250-fold, respectively [182]. These adverse effects are induced by sunlight exposure. Thiopurine derivatives absorb UVA, which is believed to be responsible for producing ^1^O_2_.

Ibuprofen and ketoprofen, both marketed as nonsteroidal anti-inflammatory drugs with antipyretic and analgesic properties, also produce ^1^O_2_ due to their photosensitizing effects and have been associated with adverse skin symptoms, including photosensitivity [183]. Some commonly used pharmaceutical compounds whose side effects include skin disorders associated with photosensitivity may also produce ^1^O_2_.

#### 5.1.8. Diagnosis and Evaluation of Skin Diseases by Measuring ^1^O_2_-Mediated Products

In skin diseases, the measurement of ^1^O_2_-derived lipid peroxidation products in sebum, especially SqOOHs, may be useful in the diagnosis of skin aging and photosensitivity. On the other hand, while the diagnosis of porphyria and SLOS can be made by measuring porphyrin and 7-DHC, respectively, the measurement of ^1^O_2_-derived lipid peroxidation products in sebum may be useful in elucidating the pathogenesis and understanding a patient’s disease status. In skin cancer, 8-oxo-dG may be useful in assessing the risk of DNA mutations induced by UV, but it should be evaluated in combination with the analysis of compounds produced by UV irradiation, such as cyclobutane pyrimidine dimers.

### 5.2. Ophthalmological Diseases

Since sunlight directly impacts the skin and eyes, oxidative stress is involved not only in the skin, but also in the eyes. In this section, we focus on eye structures, light permeation, and dysfunctions, followed by diseases related to ^1^O_2_-mediated oxidative stress.

#### 5.2.1. Structure and Optical Transparency of the Ocular Bulb

Light passes through the cornea, aqueous humor, and pupil through the lens and vitreous body, to reach the retina. The pupil regulates light, and the crystalline lens acts as the lens. The retina is a transparent membrane that lines the inside of the eyeball. It mainly consists of a layered structure of four types of cells: retinal ganglion cells, communication/glial cells, photoreceptors, and retinal pigment epithelium (RPE) cells of the vitreous. Light is received by photoreceptors; the pigment cell layer is located outside the photoreceptor layer. Under visible light, the light that passes through the eye differs depending on its wavelength, and UVA (320–400 nm), UVB (280–320 nm), and UVC (100–280 nm) are absorbed by the cornea and lens [184,185], although UV can reach the retina in an aphakic eye (i.e., an eye with no crystalline lens). In the human eye, light with a wavelength of 400–760 nm reaches the retina with little absorption from the cornea to the vitreous body. Since mid- and far-infrared rays are absorbed by water, they are absorbed by water in the cornea, lens, and aqueous humor and do not reach the fundus of the eye. Since near-infrared light has low water absorption, it penetrates the choroid [186].

#### 5.2.2. Photooxidative Stress

Visual cells, which are photoreceptors, contain visual substances of chromoproteins that receive light. They are composed of two types of cone cells that sense colors and rod cells that sense the intensity (brightness and darkness) of light. Cone cells have photopsins, and rod cells have rhodopsins. ^1^O_2_ is produced by the oxygen present in photoreceptors [187]. In addition, photoreceptor cells and retinal pigment epithelial cells accumulate waste products, called lipofuscin, with aging [188], and these are also activated by light absorption to produce radicals [189,190]. 

Lipofuscin is considered a waste product in the lysosome, consisting primarily of 30–58% protein, 19–51% lipid-like substances (oxidation products of PUFA), carbohydrates, and trace metals (2%), including iron, copper, aluminum, zinc, zinc calcium, and manganese [191]. Lipofuscin cannot be degraded by lysosomal hydrolytic enzymes because of the polymerization and cross-linking of peptides with aldehydes, resulting in a plastic-like structure that is not biologically degradable and accumulates in neurons and cardiomyocytes with age. Retinal lipofuscin has been shown to exhibit strong photosensitizing properties when photoexcited in the presence of oxygen [189]. 

Recently, fluorescent bis-retinoid N-retinyl-N-retinylidene ethanolamine (A2E) (Figure 10A,B) has been implicated in photooxidation in the retina [192,193]. A2E is the main component of lipofuscin that accumulates in RPE cells during aging. Rhodopsin, a pigment protein present in photoreceptor cells, is composed of a protein moiety called opsin and 11-*cis*-retinal (Figure 10A,B). The retina contains vitamin A (retinol). Rhodopsin is broken down into opsin and all-*trans*-retinal when it absorbs light. The RPE regenerates all-*trans*-retinal to 11-*cis*-retinal (a process called the visual cycle) (Figure 10B). During this regeneration process, all-*trans*-retinal reacts with phospholipids and another all-*trans*-retinal generates A2E (Figure 10A,B). A2E, a major component of lipofuscin, generates ^1^O_2_ upon light stimulation, especially high-energy blue light [194].

ROS and radicals generated by light act on the cell membrane lipids of photoreceptors to form lipid radicals [195,196]. Recently, photo-oxidative stress has been found to lead to retinal aging and is a factor in the onset of age-related macular degeneration [195,196].

#### 5.2.3. Compounds Protecting against ^1^O_2_ in the Macular Pigment and Lens

The macula is susceptible to oxidative damage owing to the lack of a retinal inner layer, which is affected by intense light. Therefore, the retina has a macular pigment composed of lutein and zeaxanthin, which are carotenoids, and meso-zeaxanthin [197], which is converted from lutein by the RPE-65 enzyme [198,199]. 

The lens contains lutein and vitamin C, which have a protective mechanism against blue light, with an absorption peak at 400–500 nm. In addition, these macular pigments and carotenoids have an antioxidant ability to eliminate ^1^O_2_ and are thought to maintain retinal homeostasis [198].

#### 5.2.4. Blue Light Hazard

Visible blue light (400–500 nm) from light-emitting diode (LED), mobile phones, and industrial equipment is increasingly being used, and we are susceptible to this in our daily lives. The accumulation of photo-oxidative stress caused by blue light, called blue light hazard, has been reported to cause chronic retinal changes [200,201].

Marie et al. exposed a 10 nm wide light band, range 390–520 nm, to primary retinal pigment epithelial cells treated with A2E and measured the precise action spectrum that produced the highest amount of ROS in the cells [194]. Of the spectrum of sunlight reaching the retina, blue light at 415–455 nm produced the highest amount of ROS and induced mitochondrial dysfunction. Lipofuscin and A2E accumulate in the RPE with aging, and blue light induce a photooxidative reaction that promotes cell death and angiogenesis. The need for the elderly to filter these wavelengths of light is emphasized.

Exposure to blue light (415–455 nm) for 15 h decreased activities of SOD and catalase and increased oxidized glutathione (GSSG) and ROS in A2E-treated RPE cells [194], suggesting that the balance between the oxidant and antioxidants is readily disrupted by massive blue light exposure [194]. 

#### 5.2.5. Age-Related Macular Degeneration

Age-related macular degeneration (AMD) is a disease in which the photoreceptor cells in the macula of the central fundus of the eye are damaged. There are two types: the exudative type with neovascularization generated from the choroid that feeds the retina, and the atrophic type without it. AMD is a multifactorial disease characterized by high oxygen consumption, massive radiation exposure, and high PUFA content in the retina. Photooxidative stress is presumed to be one of the causes of retinal disorders such as AMD [200]. PUFAs in the membranes are targets for ROS, which drastically increases the susceptibility of the retina to photochemical damage. In addition, as described above, A2E-containing lipofuscin, which accumulates in RPE cells with aging, is activated by light, and ^1^O_2_ is generated from abundant oxygen in the retina. In the early stages of the disease, the body’s defense mechanism acts. However, the lesion begins to progress as the body’s defense mechanisms decline with age.

#### 5.2.6. Glaucoma

Glaucoma is a progressive glaucomatous optic neuropathy that causes visual field loss and irreversible blindness [202,203,204]. The death and axon loss of retinal ganglion cells (RGCs) cause glaucomatous optic neuropathy [202]. Elevated intraocular pressure (IOP) is a primary risk factor for open-angle glaucoma (OAG) including primary OAG (POAG) [202]. In OAGs, the elevation of IOP is explained by a reduction in aqueous humor (AH) outflow at the trabecular meshwork (TM) due to qualitative and quantitative changes in the extracellular matrix in the TM [205]. Numerous reports have shown that various oxidative stressors induce RGC damage [206,207]. For example, antioxidant thioredoxins prevent glaucomatous tissue injury, specifically glutamate- and IOP-induced RGC death [208,209], indicating that oxidative stress is thought to be involved in IOP elevation and then RGC loss in POAG.

We previously demonstrated the involvement of oxidative stress in the pathogenesis of glaucoma by measuring AH and serum HODE levels derived from free radical-mediated oxidation [94,210,211], suggesting that systemic oxidation is at least partially involved in the disease. Furthermore, the levels of HODEs/LA (oxidized/parent molecules ratio, LA; linoleic acid) in AH correlated with those in serum, suggesting that ocular oxidative injury proceeds simultaneously with systemic oxidative stress [211]. The serum levels of 10- and 12-(*Z*,*E*)-HODEs/LA formed via ^1^O_2_ specific oxidation were correlated with IOP [94], which are indices of glaucoma severity derived from TM cell dysfunction. One possible process by which ^1^O_2_ is produced in the eye is type II photooxidation via a sensitizer present in the vicinity of the reaction milieu, similar to a cataract [212]. The specific region of oxidative injury has not been identified because sunlight does not reach TM cells. In addition, the pathways of the excretion and circulation of HODEs formed in the eyes remain undefined. However, our findings suggested that cerebrospinal HODE levels were well reflected in plasma levels [213,214]. Other studies have simultaneously analyzed the systemic and local redox status, suggesting that alterations in systemic oxidant and antioxidant levels reflect local redox status [215,216,217].

#### 5.2.7. Cataract

Epidemiological approaches have indicated that sun exposure is a risk factor in age-related cataracts [218,219]. UVB is mostly filtered out by the cornea and aqueous humor [220]. For this reason, UVA, which occupies nearly 95% of the UV in sunlight, has been associated with cataracts [221]. It has been proposed that chromophores with UVA-visible light absorption properties, which accumulate in the lens with age, influence the development of cataracts through the ROS generation by photosensitizing reactions [222].

The major protein components of the lens are crystallins, of which there are three main types: α-, β-, and γ-crystallin. The most abundant of these is α-crystallin, which functions to maintain lens transparency and acts as a protective chaperone for the lens [223]. As α-crystallin is not turned over, damage to the amino acids/proteins constituting α-crystallin accumulates. This leads to changes in refraction and lens opacity (cataract formation) [224].

Tryptophan metabolites (kynurenine (Figure 8G), 3-hydroxykynurenine (Figure 8G), 3-hydroxykynurenine o-β-D-glucoside, and 4-(2-amino-3-hydroxyphenyl)-4-oxobutanoic acid o-β-D glucoside) are present in significant concentrations in the lens and act as endogenous chromophores and UV-absorbing filters [225]. These endogenous chromophores absorb UV and become excited but regenerate to their original ground state without producing radicals or ^1^O_2_. These chromophores undergo deamination under physiological conditions and are covalently bound to proteins (crystallin) via Michael addition reactions [226]. As covalent modification progresses gradually with age, the lens turns yellow. Paradoxically, tryptophan derivatives, which act as UV filters, have also been shown to generate ^1^O_2_ by acting as photosensitizers when bound to proteins [226]. However, the association between cataract development and ^1^O_2_ remains undefined and has not been confirmed in our studies [94,210].

#### 5.2.8. ^1^O_2_-Derived Peroxidation Products in the Diagnosis and Evaluation of Ophthalmological Diseases

The measurement of ^1^O_2_-derived lipid peroxidation products in tissues removed during cataract and glaucoma surgery may provide assistance in elucidating the pathophysiology and in selecting the use of singlet oxygen scavengers. As our results indicate, measuring lipid oxides in circulation may also be useful for diagnosis and pathophysiological evaluation.

### 5.3. Diabetes Mellitus

#### 5.3.1. Biomarkers for Diabetes and Diabetic Complications

Diabetes is characterized by a deficiency of the secretion or action of insulin, resulting in microvasculature damage to the retina, renal glomerulus, and heart, as well as peripheral neuropathy. Diabetes can be fatal if its complications include nephritis and atherosclerosis. Type 2 diabetes mellitus (T2D) indicates elevated blood glucose levels caused by the impairment of insulin secretion and insulin resistance. Prediabetic states, including impaired fasting glucose, impaired glucose tolerance (IGT), or slightly elevated blood glucose levels, may precede T2D for the year [227,228]. Progression from prediabetes to T2D can be prevented or delayed by improving diet and increasing physical activity.

Diabetic vascular complications include nephropathy, myocardial infarction, and glaucoma. The early detection of diabetes is important to prevent complications. Several studies have been conducted on the early detection of these complications. Biomarkers of diabetic nephropathy include urinary heme oxygenase-1 (HO-1) [229] and 8-hydroxydeoxyguanosine [230,231], and circulating microRNA 130b [232]. S-glutathionylation [233] and oxidized dityrosine-containing protein [234] are considered candidate markers of vasculopathy in diabetes. Angiotensin-II, protein kinase C (PKC), and advanced glycation end products (AGEs) activate NADPH oxidase and upregulate the production of ROS, leading to cardiac dysfunction [235,236]. The metallic elements selenium [237], copper, and zinc [238] are associated with the detection of diabetes risk in pregnant women. Recently, band 3 anion transport protein, also known as anion exchanger 1 or band 3, or solute carrier family 4 member 1, was proposed for the early detection of the glycation of hemoglobin leading to AGEs [239,240,241]. To completely detect diabetes and its complications, a combination of several biomarkers is necessary.

#### 5.3.2. Diabetic Biomarkers of Lipid Peroxidation

Several studies have examined the association of oxidation products with diabetes pathology [242,243,244,245,246]. Griesser et al. revealed the interplay between lipid and protein modifications using animal models. A large cohort study showed that imbalances in the redox system contribute to the development of T2D [245]. Leinish et al. clarified that the structural and functional alterations in RNase A result from oxidation by ^1^O_2_ and ROO^•^, not solely from histidine and tyrosine cross-linking [246]. 

We recently found that 10- and 12-(*Z*,*E*)-HODE, ^1^O_2_-specific products derived from linoleic acid, significantly correlated with a risk index for impaired glucose tolerance and insulin resistance in oral glucose tolerance tests performed on healthy volunteers [92,247,248]. Free radical-specific lipid oxidation products, such as 9- and 13-(*E*,*E*)-HODE and 7β-hydroxycholesterol, and hydroxyeicosatetraenoic acids (HETEs), which are oxidation products derived from arachidonic acid, have not been detected. The process by which these ^1^O_2_-specific lipid oxidation products are produced in diabetic pathology is speculated to involve the reaction between H_2_O_2_ and ClO^−^ derived from MPO (Figure 2B). This ^1^O_2_ generation mechanism is mediated by MPO from activated phagocytes [249,250] or eosinophils peroxidase [251,252]. Several reports have shown that neutrophil-derived MPO plays an important role in diabetic vascular injury (Figure 11) [253,254,255]. The source of H_2_O_2_ in diabetic pathology is thought to be NADPH oxidase in the vascular endothelial cells [253]. MPO released from activated neutrophils is known to bind to the vessel wall for several days [256,257]. The H_2_O_2_ derived from hyperglycemia-activated NADPH oxidase may be used by MPO bound to the vascular endothelium to produce HOCl, resulting in the production of ^1^O_2_ and vascular injury. Compounds that inhibit the activity of MPO, such as hydroxamic acids, hydrazides, and azides, have potentially harmful side effects. In contrast, quercetin was recently reported to inhibit MPO-dependent HOCl production and prevent vascular endothelial injury [258]. Onyango et al. reviewed the contribution of ^1^O_2_ to insulin resistance and reported that ^1^O_2_ generates bioactive aldehydes and induces mitochondrial DNA modification and endoplasmic reticulum stress, which lead to insulin resistance [259]. 

Clinicians may be able to manage and/or advise patients regarding their food and exercise habits before the onset of diabetes, by evaluating the levels of ^1^O_2_-induced lipid peroxidation products in the near future. In any event, more information and studies are needed to determine the pathological significance of ^1^O_2_ and 10,12-(*Z*,*E*)-HODE. 

### 5.4. Bronchial Asthma

Bronchial asthma is characterized by chronic airway inflammation, reversible airflow obstruction, airway hyperresponsiveness, and structural changes in the airways, and its clinical manifestations include variable airway narrowing, cough, dyspnea, and wheezing [260]. Asthma is caused by multiple interactions between epigenetic regulation and environmental factors. Although this heterogeneity has made it difficult to define biologically distinct subgroups based on the differences in clinical phenotypes, recent advances in genetics and molecular biology have led to the pathogenetic classification of endotypes [261]. These characteristics of asthma may be helpful for understanding the pathophysiology and precision of medicine. 

More than 300 million people worldwide suffer from asthma, of which approximately 10% have severe asthma. Symptom control in severe refractory asthma is difficult to achieve despite optimal treatment with high-dose inhaled corticosteroids (ICS) and long-acting adrenergic receptor β2 agonists [262]. Airway inflammation in asthma is typically involved in the submucosal infiltration of activated Th2 lymphocytes, neutrophils, eosinophils, mast cells, and macrophages [263,264]. Although Th2 cytokines and eosinophilic inflammation are typical factors in the development of mild to moderate asthma in response to ICS, a subset of asthmatics with Th2 high-eosinophil-predominant have a refractory course despite optimal treatment [265]. In more severe cases of steroid-resistance, the cellular environment is characterized by airway inflammation induced by Th1/Th2 cytokine expression and neutrophil, with poorly reversible airflow obstruction [265,266,267]. In adults, elevated neutrophil counts in sputum are related to disease severity [268] and the persistence of symptoms [269]. These reports suggest an important role for neutrophils in asthma and their association with a more steroid-refractory subtype.

Neutrophils are recruited to the site of pathogen invasion for immune defense [270,271]. Several proinflammatory mediators, including cytokines, chemokines, and complements, contribute to the regulation of lung neutrophilic recruitment. Accumulated neutrophils release chemotactic factors that attract monocytes and/or macrophages to the infection site and, subsequently, exacerbate airway inflammation [272]. In particular, Th17/IL-17 and IL-8 are involved in the pathogenesis of steroid-resistant asthma [273,274]. Neutrophil extracellular traps (NETs) comprising neutrophil DNA, whose formation is facilitated by proinflammatory cytokines, are also thought to be implicated in the pathological condition of neutrophilic asthma [274,275,276,277]. Activated neutrophils release not only inflammatory cytokines, but also ROS and MPO, which damage airway endothelial cells and exacerbate allergic inflammation [278,279,280,281]. Plasma MPO levels in asthmatic patients have been proposed as biomarkers for the evaluation of asthma severity in adults [277,282]. As described in Section 2 and Section 5.3.2, MPO contributes to the production of ^1^O_2_ mediated by the progression of the reaction with H_2_O_2_ and Cl^−^ producing HClO. These suggest that ^1^O_2_ may be associated with pathogenesis phenotypes in refractory asthma. We have demonstrated a positive correlation between the levels of ^1^O_2_-mediated oxidation products 10- and 12-(*Z*,*E*)-HODEs, and the levels of MPO activity and IL-17-derived nerve growth factor (NGF) in the bronchoalveolar lavage fluid (BALF) of an asthmatic mouse model with mixed inflammation [97]. As increased NGF in the BALF of asthmatic patients induces smooth muscle hyperplasia in the airway [283,284] and anti-NGF antibodies improve airway hyperresponsiveness [285], NGF is also considered a therapeutic target for lung disease [286]. Interestingly, the generation of ^1^O_2_ with an endoperoxide [(3-(1, 4-epidioxy-4-methyl-1, 4-dihydro-1- naphthyl) propionic acid] increased NGF and IL-8 levels in human bronchial epithelial cells, suggesting that ^1^O_2_ generation may be an upstream event of increase in NGF.

Recent studies have described neutrophil-derived MPO as a potential biomarker for neutrophilic asthma [277,282]. Therefore, we would expect that 10, 12-(*Z*,*E*)-HODEs, which are elevated according to neutrophil recruitment, could also be a prominent biomarker with more stability than MPO for neutrophilic asthma. Glucocorticoids, which are used as the first choice for asthma therapy, fail to attenuate neutrophilic inflammation and may even promote neutrophil survival [268,287]. It seems reasonable to choose macrolide antibiotics for the treatment of asthma rather than steroids when the levels of 10- and 12-(*Z*,*E*)-HODEs are increased. However, the use of macrolides is not appropriate for the long-term treatment of neutrophilic asthma due to the risk of bacterial resistance. Although future studies are needed to determine how ^1^O_2_ is related to the pathological condition of asthma patients, 10- and 12-(*Z*,*E*)-HODEs may be useful in stratifying patients with refractory asthma and in selecting treatment options.

## 6. ^1^O_2_ Scavengers

The living body contains superoxide dismutase and catalase, which can scavenge ROS such as O_2_^•−^ and H_2_O_2_, but it does not have enzymes that can scavenge ^1^O_2_. Substances that can effectively scavenge ^1^O_2_ depend on exogenous compounds such as dietary carotenoids.

### 6.1. Carotenoids

Carotenoids are plant pigments that are pro-vitamin A and potent scavengers of ^1^O_2_. The ^1^O_2_ scavenging of carotenoids is mainly based on physical scavenging. The excitation energy of ^1^O_2_ is transferred to carotenoids (singlet carotenoids; ^1^Car) to produce triplet oxygen (^3^O_2_) and triplet carotenoids (^3^Car). ^3^Car, which receives excitation energy, returns to ground state carotenoids by releasing heat as energy. Thus, carotenoids can repeatedly scavenge ^1^O_2_ [288].
^1^Car + ^1^O_2_ → ^3^Car + ^3^O_2_

Carotenoids are the most potent ^1^O_2_ scavengers found in nature, and carotenoids with 11 carbon atoms involved in the π-conjugation length, such as lycopene (Figure 12A), β-carotene (Figure 12B), and astaxanthin (Figure 11C), are the most efficient ^1^O_2_ scavengers [288]. Among carotenoids, studies have mainly been conducted on the products associated with the ^1^O_2_ scavenging of β-carotene, and β-carotene 5,8-endoperoxide (Figure 12B) is a specific product mediated by ^1^O_2_ oxidation [102,289,290]. In contrast, β-carotene 5,6-epoxide is formed by a free radical-mediated reaction. β-Carotene-5,8-endoperoxide has been detected in vitro [291,292] and in vivo [289]. Among carotenoids, lycopene has the strongest ^1^O_2_ scavenging capacity [293]. The second-order rate constants of physical quenching (kq) in ethanol/chloroform were 31,000 × 10^6^ M^−1^s^−1^ for lycopene, 14,000 × 10^6^ M^−1^s^−1^ for β-carotene, 8000 × 10^6^ M^−1^s^−1^ for lutein, and 10,000 × 10^6^ M^−1^s^−1^ for zeaxanthin, indicating that lycopene was the most efficient ^1^O_2_ quencher [294]. Lycopene and β-carotene also exhibited the fastest ^1^O_2_ scavenging rate constants under the conditions of the model membrane system using liposomes [295]. Unlike other carotenoids, lycopene does not possess a ring structure. Since 2-methyl-2-hepten-6-one (Figure 12A) and apo-6′-lycopenal (Figure 12A) were detected under 500 W light irradiation in the presence of methylene blue, a photosensitizer, lycopene is assumed to react with ^1^O_2_ via a dioxetane intermediate [296,297].

Carotenoids in plants exist in close proximity to chlorophyll in chloroplasts and scavenge ^1^O_2_ generated by the photosensitizing effect of chlorophyll during light absorption. Thus, carotenoids have a protective property against light stress in plants. In addition, carotenoids, such as xanthophylls (lutein (Figure 12D) and zeaxanthin (Figure 12E), specifically accumulate in the macula of the retina, which is an organ exposed to light. In sunlight, the energy of blue light (400 nm), in particular, is 100-fold greater than that of red light (590 nm) and causes severe damage to cells. Since lutein and zeaxanthin have absorption maxima for light at approximately 440 nm, these carotenoids can efficiently absorb blue light. Therefore, the accumulation of lutein and zeaxanthin in the retinal macula can prevent the oxidative degeneration of macular tissue and reduce the risk of AMD and other diseases. Lutein and zeaxanthin are also present in the lens, where they prevent the development of cataract [298,299]. The consumption of tomatoes rich in β-carotene and lycopene can reduce the formation of erythema in human skin due to UV irradiation [300,301]. 

### 6.2. Vitamin E (Tocopherol)

α-Tocopherol, a primary fat-soluble antioxidant in vivo, also has ^1^O_2_ scavenging capacity that is approximately 30–100 times weaker than that of carotenoids [302]. However, the ^1^O_2_ scavenging capacity of carotenoids is also reported to be reduced in liposomes, which are biomembrane models, compared to solution [303].Considering the level of tocopherols in vivo, the in vivo ^1^O_2_ scavenging activity of α-tocopherol is considered to be functional. Among tocopherols, α->β->γ->δ-tocopherol have ^1^O_2_ capacities in this order [304,305]. Tocotrienols, which are homologues of tocopherols, also have ^1^O_2_ scavenging ability in the order of α->β->γ->δ-tocotrienol [304]. 

The oxidation product of α-tocopherol produced by ^1^O_2_ is α-tocopherylquinone, which can be measured in biological samples [306,307,308]. α-Tocopherylquinone is not an ^1^O_2_-specific marker because it is also produced by ROS other than ^1^O_2_.

### 6.3. Other Compounds

As previously mentioned, PUFA is susceptible to oxidation by ^1^O_2_ and has ^1^O_2_ scavenging capacity, although it is very weak compared to carotenoids [309].

Bakuchiol (Figure 13A), a terpeno-phenolic compound, is a functional analog of retinol that has attracted attention in skincare for its ability to induce retinol gene expression and stimulate collagen production [310]. Bakuchiol protects retinol from degradation by ^1^O_2_ produced by H_2_O_2_ and lithium molybdenum. It was shown to inhibit the ^1^O_2_-induced peroxidation of squalene, prevent pore clogging, and reduce the onset of acne by 42% with topical bakuchiol treatment in 54 volunteers [311]. In addition, the 12-week application of a cream containing bakuchiol (0.5%) improved wrinkles and pigmentation [312]. 

Zingerone (4-[4-hydroxy-3-methoxyphenyl]-2-butanone) from *Zingiber officinale Roscoe* (ginger) scavenges ^1^O_2_ [313]. The structure of zingerone is similar to that of turmeric-derived curcumin (1,7-bis[4-hydroxy-3-methoxyphenyl]-1,6-heptane-3,5-dione). Zingerone and curcumin can both prevent skin aging. Acetyl zingerone (3-(4-hydroxy-3-methoxybenzyl)pentane-2,4-dione) (Figure 13B), a multifunctional skincare ingredient, was synthesized using the molecular structures of zingerone and curcumin [314]. Curcumin has been shown to have ^1^O_2_ scavenging activity in an in vitro experimental system by ESR spectroscopy using TEMP as a spin-trap [315]. Acetyl zingerone, which has a higher ^1^O_2_ scavenging activity than α-tocopherol [316], decreases the expression of genes associated with extracellular matrix degradation (MMP-3 and cathepsin V) and inhibits the activity of MMP-1, -3, and -12 [314]. Clinical reports have demonstrated that treatment with a cream containing 1% acetyl zingerone applied twice daily for 8 weeks improves wrinkles, hyperpigmentation, and erythema [317].

## 7. Summary and Outlook

As previously described, diseases and dysfunctions mediated by or, at least, related to ^1^O_2_ can be categorized into two groups: those caused by the UV irradiation of photosensitizers accumulated in the living body, and those resulting from the activation of MPO in leukocytes. The former includes cutaneous photosensitivity and retinal diseases, which have been clinically investigated. For the latter, we have previously reported findings in diabetic patients, glaucoma patients, and in an asthma mouse model. MPO is expressed in leukocytes and is implicated in many inflammatory diseases [318]. MPO is associated with cardiovascular disease, atherosclerosis, glomerulonephritis, arthritis, and Alzheimer’s disease. Practical biomarkers are required for the early diagnosis, assessment, and progression of diseases, and for the evaluation of treatment efficacy. However, as summarized in Table 2 and Table 3, there are few reports on the analysis of lipid peroxidation products formed by ^1^O_2_ in clinical samples, except for SqOOHs. SqOOHs can be measured noninvasively as they are found in sebum; therefore, an increasing number of reports are available (Table 3), especially in the cosmetics and beauty industries. SqOOHs are lipid peroxidation products, but not specific products of ^1^O_2_. Since lipid peroxidation is a major oxidative injury in vivo, lipid peroxidation products, especially hydroxides, may be useful biomarkers. It should be noted, however, that lipid peroxidation products are generally not diagnostic indicators of specific diseases. Their usefulness should be enhanced by combination with other biomarkers. Further prospective studies and analysis of the correlation between lipid peroxidation products, specifically from ^1^O_2_, and the severity of disease progression should be conducted in the future. When disease risk can be assessed by a prominent biomarker, advice on diet, including carotenoids, and exercise habits can be provided.

## Figures and Tables

**Figure 1 ijms-24-02739-f001:**
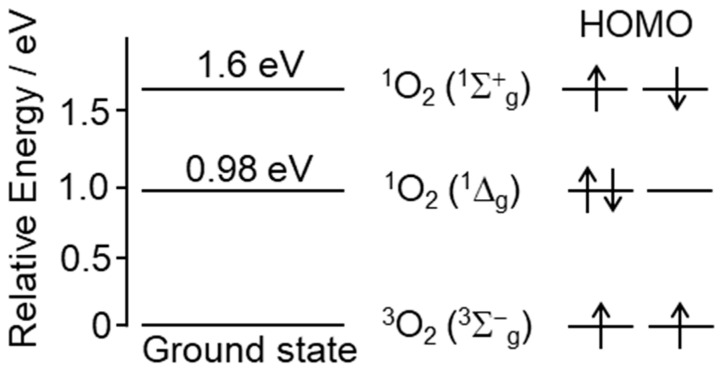
Relationship between energy levels of ground state triplet oxygen molecule ^3^O_2_ and singlet oxygen molecule ^1^O_2_**.** The ground state oxygen molecule, triplet oxygen ^3^O_2_ (^1^Σ^−^_g_), has two unpaired and spin-parallel electrons in π* antibonding orbitals. ^3^O_2_ (^3^Σ^−^_g_) in the ground state receives energy and is excited to the singlet state, forming singlet oxygen (^1^O_2_). ^1^O_2_ is the spin-flipped electron state of ^3^O_2_ (^3^Σ^−^_g_). There are two states of ^1^O_2_:^1^O_2_ (^1^Σ^+^_g_) and ^1^O_2_ (^1^Δ_g_). ^1^O_2_ (^1^Σ^+^_g_) and ^1^O_2_ (^1^Δ_g_) have energies that are 1.6 eV and 0.98 eV higher than the ground state ^3^O_2_ (^3^Σ^−^_g_), respectively. The lifetime of ^1^O_2_ (^1^Σ^+^_g_) is a few picoseconds, and it is rapidly converted to ^1^O_2_ (^1^Δ_g_). Since the lifetime of ^1^O_2_ (^1^Δ_g_) is several microseconds, which is significantly longer than that of ^1^O_2_ (^1^Σ^+^_g_), the ^1^O_2_ generated in vivo can be ^1^O_2_ (^1^Δ_g_). The arrows indicate the direction of the electron spin of the highest occupied molecular orbital (HOMO).

**Figure 2 ijms-24-02739-f002:**
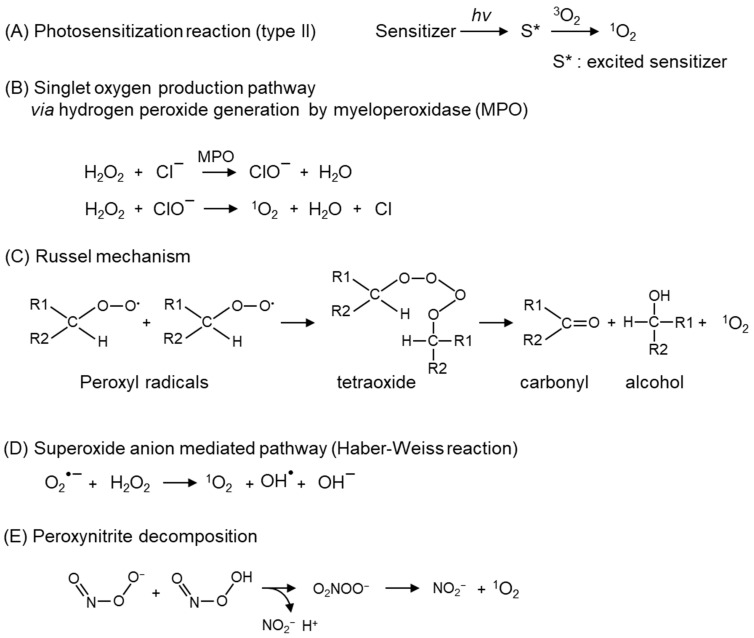
Major ^1^O_2_ production mechanisms. ^1^O_2_ production pathways; (**A**) photochemical reaction, (**B**) reactions mediated by hydrogen peroxide produced by myeloperoxidase (MPO), (**C**) decomposition of peroxyl radicals (Russel mechanism), (**D**) reaction mediated by superoxide anion (O_2_^•−^), (**E**) pathway via peroxynitrite decomposition.

**Figure 3 ijms-24-02739-f003:**
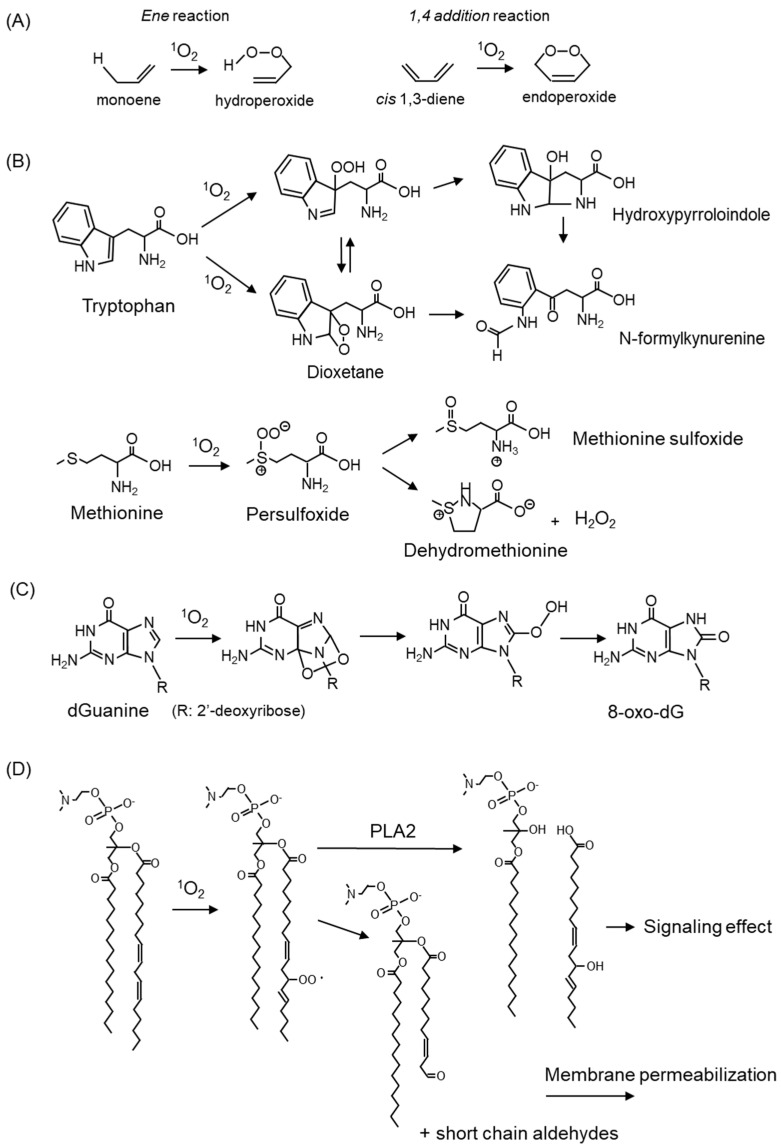
^1^O_2_-mediated oxidation of amino acid, nucleic acid, and lipids. (**A**) Oxygenation reaction by ^1^O_2_. (**B**) Reaction products of tryptophan and methionine with ^1^O_2_. (**C**) Formation of 8-oxo-2′-deoxyguanosine (8-oxo-dG) by ^1^O_2_ oxidation to deoxyguanosine (dG). (**D**) Oxidative modification of fatty acids in membrane phospholipids by ^1^O_2_.

**Figure 4 ijms-24-02739-f004:**
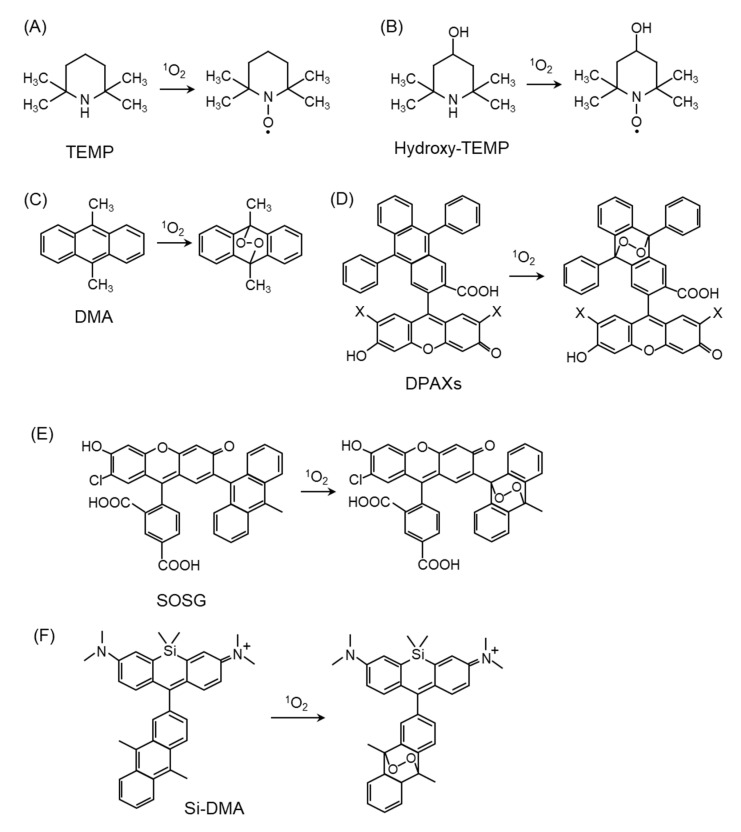
Structures and oxidation modification of probes for ^1^O_2_ detection. (**A**) 2,2,6,6-tetramethylpiperidine (TEMP), (**B**) Hydroxy-TEMP, (**C**) 9,10-dimethylanthracene (DMA), (**D**) 9-[2-(3-Carboxy-9,10-diphenyl)anthryl]-6-hydroxy-3H-xanthen-3-ones (DPAXs), (**E**) Singlet Oxygen Sensor Green (SOSG), (**F**) silicone-containing rhodamine-9,10-dimethylanthracene (Si-DMA).

**Figure 5 ijms-24-02739-f005:**
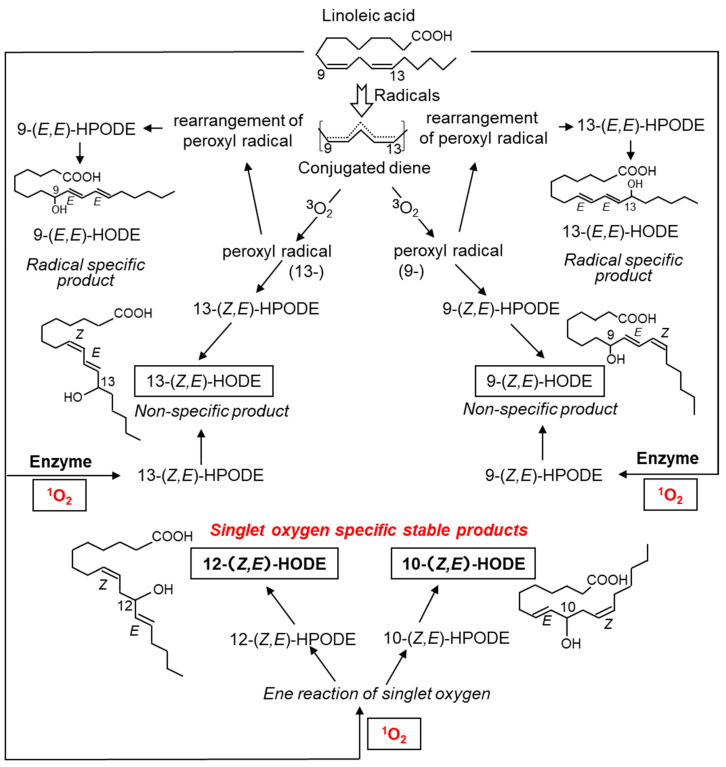
Structures and formation mechanism of hydroxyoctadecadienoic acid (HODE), an oxidation product derived from linoleic acid. ^1^O_2_ oxidizes linoleic acid to produce 13-(*Z*,*E*)-HODE, 9-(*Z*,*E*)-HODE, 12-(*Z*,*E*)-HODE, and 10-(*Z*,*E*)-HODE. 13-(*Z*,*E*)-HODE and 9-(*Z*,*E*)-HODE are also produced by ROS other than ^1^O_2_ and by enzymatic oxidation reactions via lipid oxidases. 9-(*E*,*E*)-HODE and 13-(*E*,*E*)-HODE are produced in a radical-specific manner.

**Figure 6 ijms-24-02739-f006:**
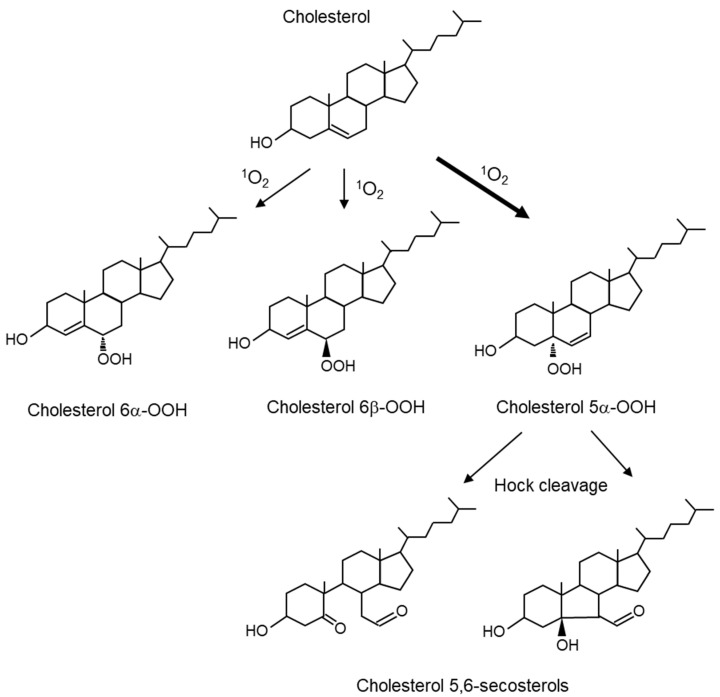
Structures of cholesterol peroxides produced by ^1^O_2_-mediated oxidation reactions. ^1^O_2_ oxidizes cholesterol to produce 5α-hydroperoxide (cholesterol 5α-OOH), 6α-hydroperoxide (cholesterol 6α-OOH), and 6β-hydroperoxide (cholesterol 6β-OOH). Hock cleavage of cholesterol 5α-OOH converts it to cholesterol 5,6-secosterols.

**Figure 7 ijms-24-02739-f007:**
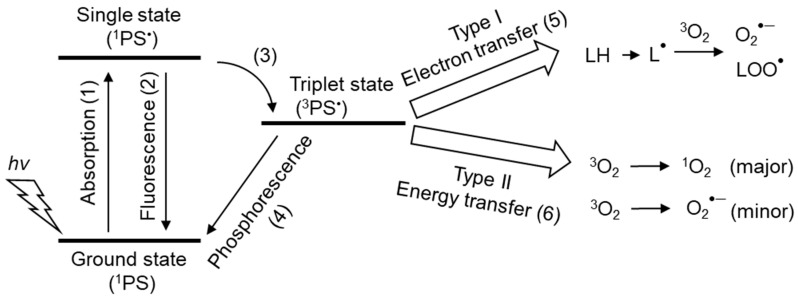
Photosensitizers and the mechanisms of ^1^O_2_ production. (1) Photosensitizers (PS) absorb photons (*hv*) from light and transform them to the excited singlet state (^1^PS^•^). (2) ^1^PS^•^ returns to the ground singlet state (^1^PS) by losing energy via fluorescence emission. (3) ^1^PS^•^ was converted to the long-lived triplet state (^3^PS^•^). (4) ^3^PS^•^ can return to the ground singlet state (^1^PS) via light emission (phosphorescence). (5) ^3^PS^•^ forms organic radicals (L^•^) from organic compounds (LH) through the transfer of electrons in the type I photochemical reaction. L^•^ affects oxygen (^3^O_2_) to produce ROS (superoxide anion O_2_^•−^, lipid peroxyl radical (LOO^•^). O_2_^•−^ leads to the production of H_2_O_2_ and hydroxyl radicals (OH^•^), resulting in the induction of radical chain reactions. (6) In type II photochemical reactions, the energy of ^3^PS^•^ is transferred to ^3^O_2_ to mainly produce ^1^O_2_, but O_2_^•−^ is produced as a minor product via electron transfer from ^3^PS^•^.

**Figure 8 ijms-24-02739-f008:**
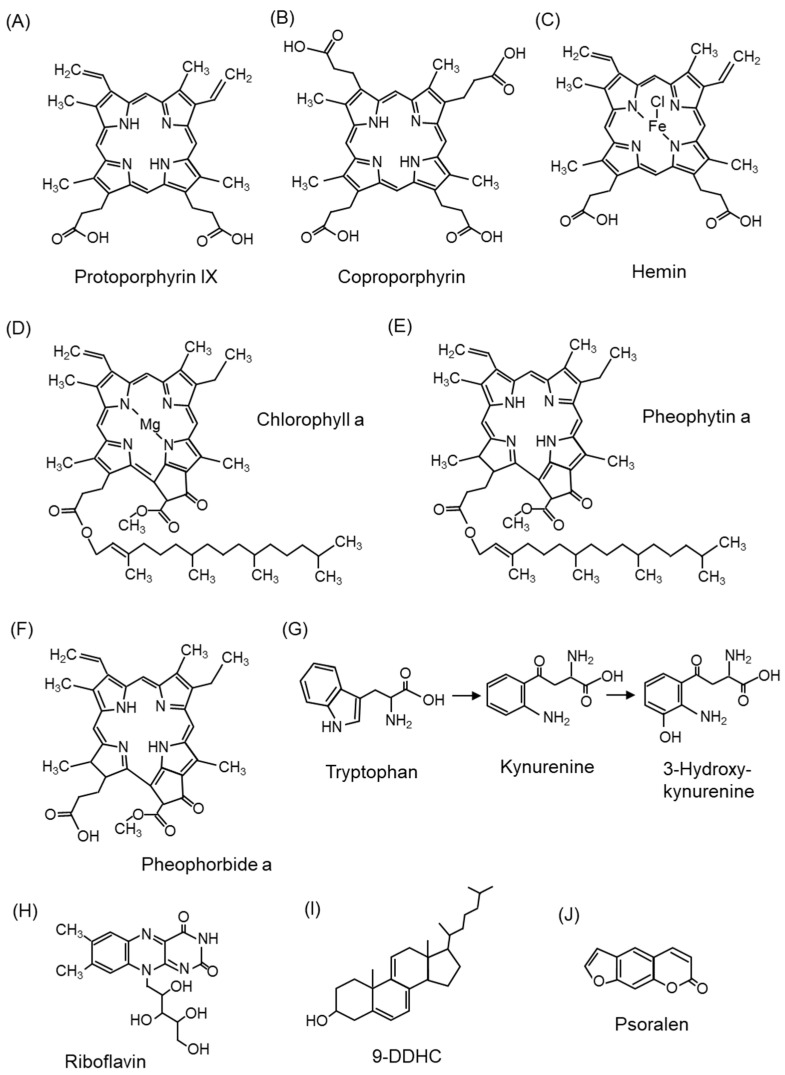
The structures of photosensitizers. (**A**) Protoporphyrin IX, (**B**) coproporphyrin, (**C**) hemin, (**D**) chlorophyll a, (**E**) pheophytin a, (**F**) pheophorbide a, (**G**) tryptophan, (**H**) riboflavin, (**I**) cholesta-5,7,9(11)-trien-3beta-ol (9-DDHC), (**J**) psoralen.

**Figure 9 ijms-24-02739-f009:**
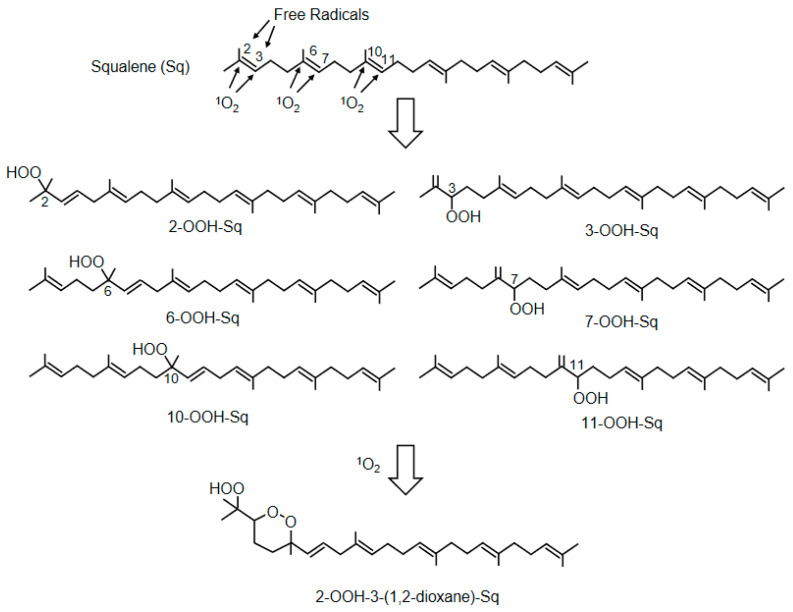
The structures of squalene and its peroxidation products. Squalene (Sq) is converted by ^1^O_2_ to six monohydroperoxides (2-, 3-, 6-, 7-, 10-, and 11-OOH-Sq). The monohydroperoxides of Sq are further photo-oxidized to 2-OOH-3-(1,2-dioxane)-Sq.

**Figure 10 ijms-24-02739-f010:**
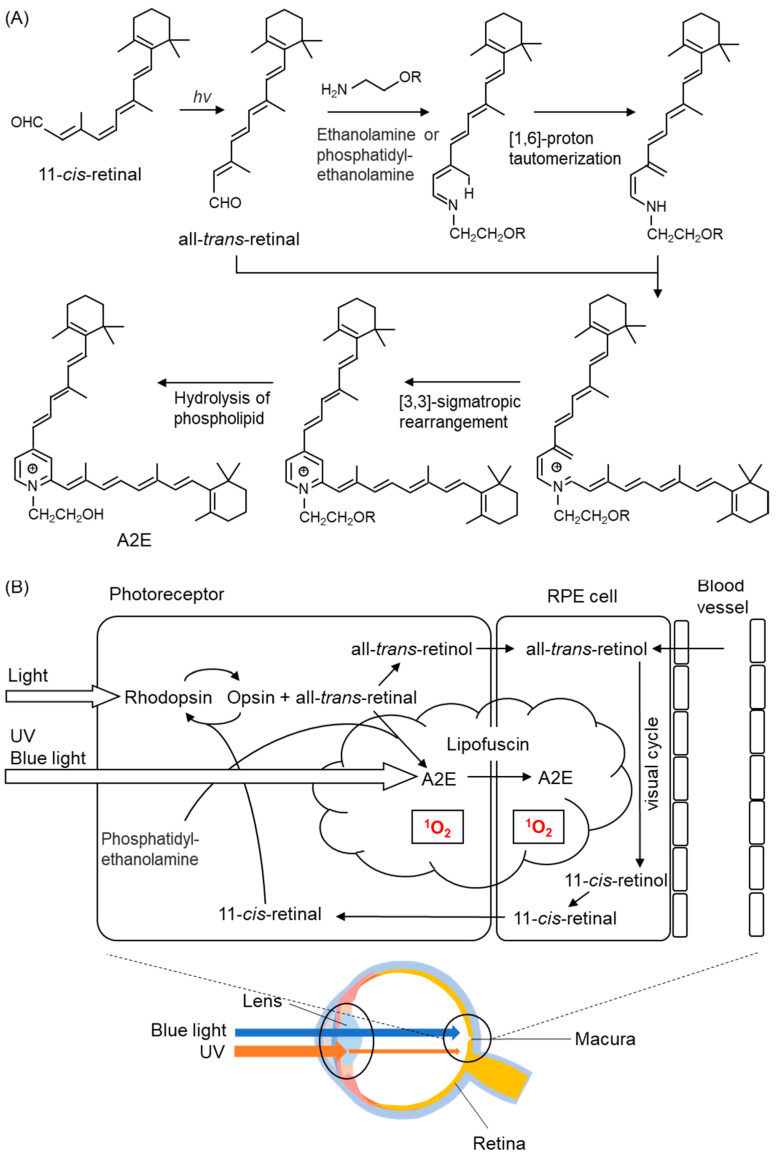
The structures of biosynthesis of A2E. (**A**) After Schiff base formation between all-*trans*-retinal and ethanolamine or phosphatidylethanolamine, a [1,6]-proton tautomerization to enamine follows. After further Schiff base formation with a second molecule of all-*trans-*retinal, a [3,3]-sigmatropic rearrangement is followed by the hydrolysis of the linked phosphatidylethanolamine adduct, resulting in the formation of N-retinyl-N-retinylidene ethanolamine (A2E). (**B**) ^1^O_2_ production process mediated by A2E in the retina. Light irradiation of A2E accumulated in lipofuscin generates ^1^O_2_.

**Figure 11 ijms-24-02739-f011:**
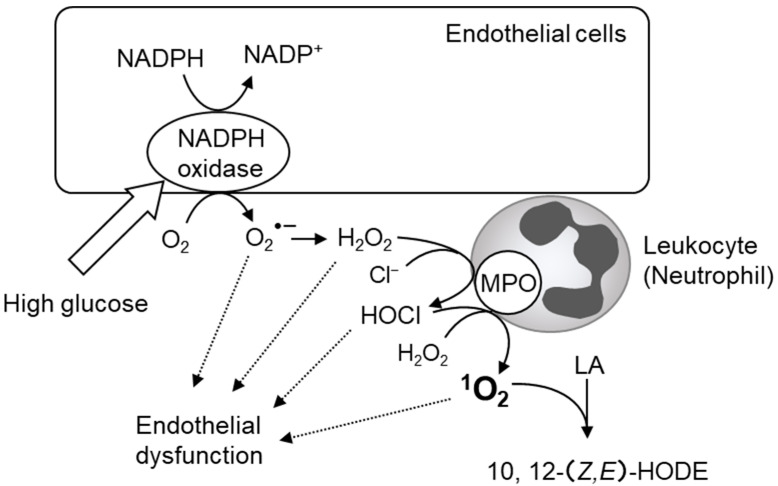
Assumed scheme of ^1^O_2_ production via NADPH oxidase and myeloperoxidase (MPO) in vascular injury in diabetes mellitus. High glucose produces hydrogen peroxide (H_2_O_2_) by NADPH oxidase in vascular endothelial cells. ^1^O_2_ is generated via MPO in activated neutrophils bound to the vessel wall.

**Figure 12 ijms-24-02739-f012:**
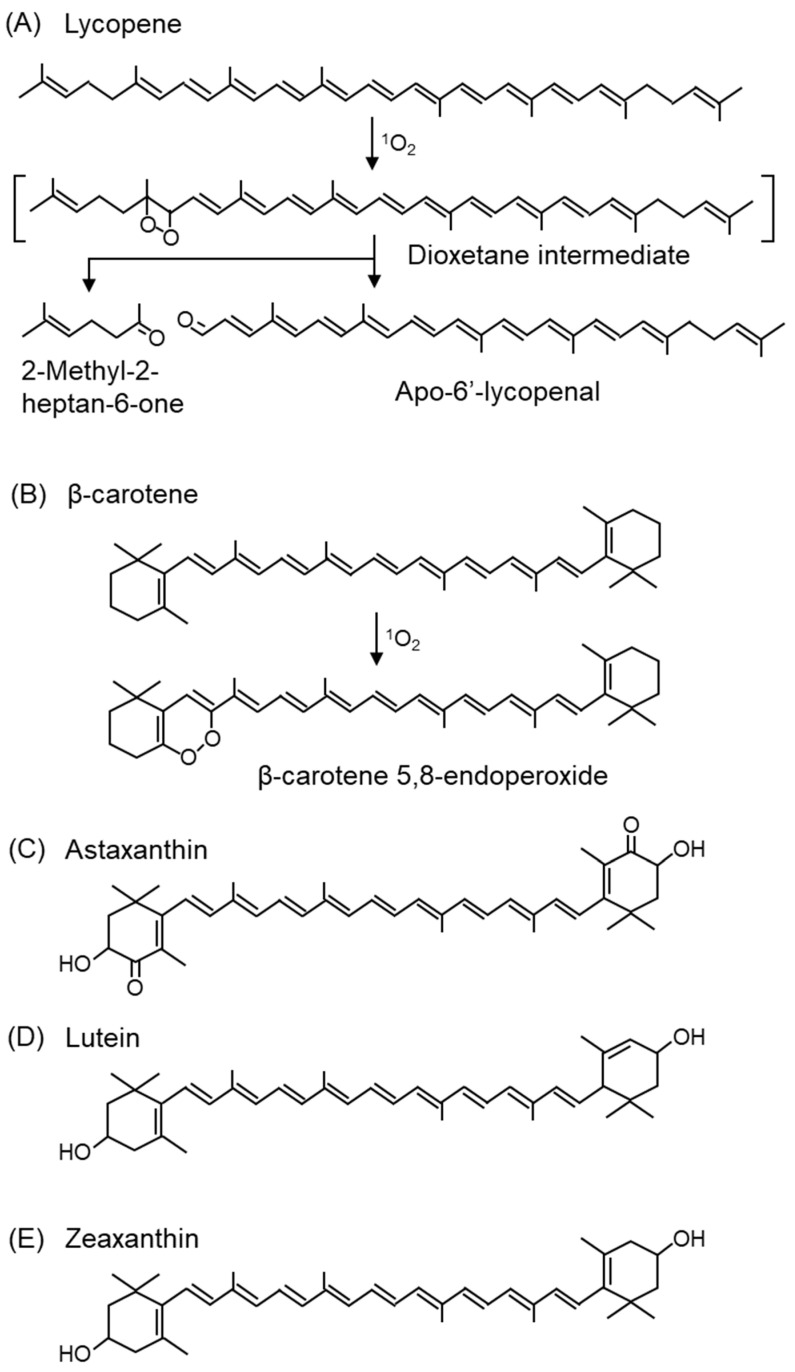
The structures of carotenoid and peroxidation products from lycopene and β-carotene. (**A**) Lycopene, (**B**) β-carotene, (**C**) astaxanthin, (**D**) lutein, (**E**) zeaxanthin.

**Figure 13 ijms-24-02739-f013:**
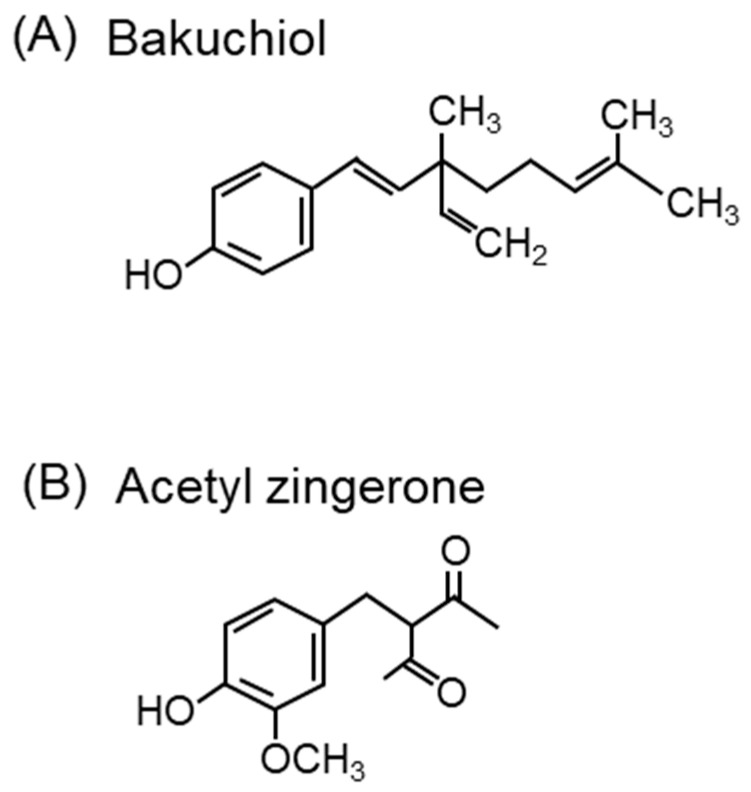
Compounds with ^1^O_2_ scavenging activity other than carotenoids. The structures of (**A**) bakuchiol and (**B**) acetyl zingerone.

**Table 1 ijms-24-02739-t001:** Characterization of ^1^O_2_ detection methods in vivo and in vitro.

Detection Method of ^1^O_2_	Principle	Pros/Cons	Ref.
Near-infrared luminescence	^1^O_2_ emission at 1270 nm	A standard technique for the ^1^O_2_ yields, lifetimes, and deactivation constants in vitro. Weak signal (phosphorescence yields are of the order 10^−5^ to 10^−7^) Requirement of deuterium oxide (D_2_O) for longer lifetime ➢ Applied for in vitro detection	[54,57,58,59]
ESR spectroscopy	Measurement of nitroxide radicals produced by the reaction of ^1^O_2_ with sterically hindered secondary amine probes	Applied for in vitro ^1^O_2_ scavenging activity in solution Time resolution is longer for the short lifetime of ^1^O_2_ in cells ➢ Applied almost for in vitro detection	[48,60,61,65,66]
Fluorescent probes			[67,68,69]
9,10-dimethylanthracene (DMA)	React with ^1^O_2_ and form stable endoperoxides with high-fluorescence quantum yield (high sensitivity, simplicity of data acquisition, and high spatial resolution in microscopic imaging)	Detection of ^1^O_2_ in neutral or basic aqueous solutions React rapidly with ^1^O_2_ in a high rate constant Impermeable cell membrane ➢ Applied for in vitro detection	[68,70,71,72]
9-[2-(3-Carboxy-9,10-diphenyl)anthryl]-6-hydroxy-3H-xanthen-3-ones (DPAXs)			
Singlet Oxygen Sensor Green (SOSG)		Commercially available highly selective ^1^O_2_ indicator Impermeable cell membrane ➢ Applied for in vitro detection	[73,74,75]
SOSG-based nanosensors NanoSOSG SPR-SOSG PAM-SOSG		Using nanoparticles for detection of intracellular ^1^O_2_ Permeable cell membrane Visualization of ^1^O_2_ signal at the subcellular level ➢ Applied for in vitro and intracellular detection	[50,78,79]
Si-DMA		Cell-permeable and localization of mitochondria Specifically detect mitochondrial ^1^O_2_ Relatively quantitative ➢ Applied for intracellular detection	[76,77]
^1^O_2_-mediated peroxidation products			
10- and 12-(*Z*,*E*)-HODEs	Products mediated by the reaction of ^1^O_2_ with linoleic acid	Relatively stable Abundant in living organisms (cell membrane) Can be used to evaluate in vivo oxidative damage ➢ Applied for in vivo and in vitro detection	[83,89,90,91,92,93]
Cholesterol 5α-OOH Cholesterol 6α-OOH Cholesterol 6β-OOH	Products mediated by the reaction of ^1^O_2_ with cholesterol	Relatively stable Can be used to evaluate in vivo oxidative damage ➢ Applied for in vivo and in vitro detection	[52]

**Table 2 ijms-24-02739-t002:** Report on the results of the determination of linoleic acid- or cholesterol-derived peroxides produced by ^1^O_2_-mediated oxidation reactions using human and animal samples.

^1^O_2_-Mediated Oxidation Product	Sample Species	Disease/Model	Ref.
10- and/or 12-(*Z*,*E*)-HODE	Human	Borderline diabetes/oral glucose tolerance test	[92,93]
	Human	High serum levels in patients with primary open-angle glaucoma	[94]
	Pig	Increased by kidney tissue damage	[95]
	Mouse	High in the plasma of model mice with prediabetes	[83,96]
	Mouse	Increased in the lungs of model mice with asthma and positively correlated with MPO activity and nerve growth factor (NGF)	[97]
Cholesterol 5α-hydroperoxide	Human	Alcoholic patients	[98]
	Rat	Pheophorubide a and visible light irradiation	[99]
	Mouse	UVA irradiation of hairless mice	[100]
	Mouse	Effect of beta-carotene on UVA irradiation of hairless mice	[101,102]
Cholesterol 5,6-secosterol	Human	Cholesterol 5,6-secosterol detected in atherosclerotic plaques	[103]
	Human	Cortex in patients with Alzheimer’s disease	[104]
	Rat	Higher in the plasma of amyotrophic lateral sclerosis (ALS) rats than before disease onset	[85]

## Data Availability

Not applicable.

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
