# Peer review of "Significance of Singlet Oxygen Molecule in Pathologies"

_ijms, 2023, doi:10.3390/ijms24032739_

Round 1

Reviewer 1 Report

The review article entitled Significance of singlet oxygen molecule in pathologies submitted to IJMS by Murotomi and co-workers have examined the reactive oxygen species, including singlet oxygen, in the progression of disease as well as in aging. In particular the involvement of singlet oxygen in the pathogenesis of skin and eye diseases is discussed from the biomolecular perspective. The authors referred lipid oxidation products derived from singlet oxygen-mediated oxidation in glaucoma, early diabetes patients, and a mouse model of bronchial asthma; Recent developments in singlet oxygen scavengers such as carotenoids, which can be utilized to prevent the onset and progression of disease, have also been described. The manuscript focuses an interesting topic that is worth to be published, but after minor revision.

-Moderate English changes are required

-In general, I suggest to review the style of the manuscript according to the guidelines of the journal.

In the paragraph 5.3. Diabetes mellitus

5.3.1. Biomarkers for diabetes and diabetic complications

The authors focused on an interesting topic, because diabetes is one of the most common diseases in the world and the identification of new predictive biomarkers of the disease is constantly growing. In this regard, I suggest introducing a recent study in which the measurement of the anion exchange capacity of Band 3 protein, highly abundant on erythrocytes, is performed as a suitable tool to monitor the impact of hyperglycemia on erythrocyte homeostasis, being the first line of high impact of glucose before glycation of Hb, which reveals the production of AGEs. In this regard, I suggest you consider the following bibliography:

https://doi.org/10.3390/antiox9050365

https://doi.org/10.1371/journal.pone.0129495

Reviewer 2 Report

In this study, Murotomi et al describe the molecular mechanism of singlet oxygen production, its role in skin and eye-associated pathogenesis, and methods for evaluation of singlet oxygen-induced damages. Moreover, biological consequences associated with singlet oxygen-mediated oxidation in glaucoma, diabetes and asthma were also discussed. Overall, the manuscript is well written, though it requires minor grammatical error checks. Their observations are very interesting, comprehensive and summarize significant findings in this review. 

Reviewer 3 Report

In the manuscript: “Significance of singlet oxygen molecule in pathologies.” The authors reviewed the involvement of singlet oxygen in pathogenesis. Also, they discussed its biomolecular mechanisms. An interesting topic that contributes to knowledge in the area, but certain issues must be corrected.

Major revisions

1.    In the abstract and introduction, the objective of the manuscript must be mentioned, mentioning the gap that the manuscript will fill within the current knowledge, the topics that will be reviewed in the manuscript, and the possible conclusions that the reader will find through the manuscript.

2.    Please, the authors must put a reference to lines 55-64.

3.    Please, the authors must put a reference to lines 78-86. Also, please mention specifically what the authors mean in the following statement " allowing it to diffuse over a broader area in living organisms and cause damage through reacting with biomolecules" if the authors mean that 1O2 diffuses into the cell through different organelles, mention which cell organelles it diffuses depending on where it is produced; for example, if it is produced in the mitochondria, can 1O2 diffuse towards the cell cytoplasm? or vice versa, if it occurs in the cytosol, is it possible that it diffuses towards the mitochondria or other organelles?

4.    In figures 2, 5, 6, 10, 11, and 13, the authors must add a description of the figure legend.

5.    The authors mention tables 1 and 2, but the tables are not shown in the manuscript.

6.    The authors must make a table that shows the methods to be able to quantify the 1O2. This table would help quickly to identify the methods and their pros or cons for the identification of 1O2.

7.    The sections of the manuscript must be improved by adding a conclusion at the end of each section, mentioning why this assay in question is better or not to measure 1O2 or mentioning the relevance of 1O2 in that disease.

8.    The section: “5.1.4. Skin cancer” must be amplified since the relevance of 1O2 is not mentioned. The section deals with general mechanisms associated with mutation and does not specify the relevance of 1O2 in skin cancer.

9.    the authors mention curcumin in section: "6.3. Other Compounds"; however, the mechanism by which it could be protective against 1O2 is not explored.

10. The authors must improve the quality of their images, especially in figure 13, since it is pixeled.

Minor revisions

1.    Define all abbreviations to be presented in the text. For instance, NADPH is not defined.

Round 2

Reviewer 3 Report

The authors have resolved the reviews appropriately.